



# Country resolved combined emission and socio-economic pathways based on the RCP and SSP scenarios

Johannes Gütschow[1], M. Louise Jeffery[1,3], Annika Günther[1], and Malte Meinshausen[2]

[1]Potsdam Institute for Climate Impact Research, Telegraphenberg, Potsdam, Germany
[2]Climate & Energy College, School of Earth Sciences, The University of Melbourne, Australia
[3]NewClimate Institute, Berlin, Germany

**Correspondence:** johannes.guetschow (at) pik-potsdam.de

**Abstract.** Climate policy analysis needs reference scenarios to assess emissions targets and current trends. When presenting their national climate policies, countries often showcase their target trajectories against fictitious so-called baselines. These counterfactual scenarios are meant to present future Greenhouse Gas (GHG) emissions in the absence of climate policy. These so-called baselines presented by countries are often of limited use as they can be exaggerated and the methodology used to
derive them is usually not transparent. Scenarios created by independent modeling groups using integrated assessment models (IAMs) can provide different interpretations of several socio-economic storylines and can provide a more realistic backdrop against which the projected target emission trajectory can be assessed. However, the IAMs are limited in regional resolution. This resolution is further reduced in intercomparison studies as data for a common set of regions are produced by aggregating the underlying smaller regions. Thus, the data are not readily available for country-specific policy analysis. This gap is closed
by downscaling regional IAM scenarios to country-level. The last of such efforts has been performed for the SRES scenarios (Special Report on Emissions Scenarios), which are over a decade old by now. CMIP6 scenarios have been downscaled to a grid, however they cover only a few combinations of forcing levels and SSP storylines with only a single model per combination. Here, we provide up to date country scenarios, downscaled from the full RCP (Representative Concentration Pathways) and SSP (Shared Socio-Economic Pathways) scenario databases, using results from the SSP GDP (Gross Domestic Product)
country model results as drivers for the downscaling process. The data is available at https://doi.org/10.5281/zenodo.3638137 (Gütschow et al., 2020).

## 1 Introduction

In order to coordinate climate change research, different sets of joint scenarios have been developed. For example, the Intergovernmental Panel on Climate Change (IPCC) Special Report on Emission Scenarios (SRES) summarized available literature
and provided six illustrative marker scenarios of emissions as well as socio-economic storylines to enable cross-comparison of a wide range of mitigation, adaptation, and climate change impact studies (Nakicenovic and Swart, 2000; Riahi et al., 2007). These "marker scenarios are no more or less likely than any other scenarios, but are considered by the SRES writing team as illustrative of a particular storyline" (Nakicenovic et al., 2000). More recently a new scenario process was started (Moss et al., 2010). The emissions scenarios used in that process are the Representative Concentration Pathways (RCPs) that have been



developed on the basis of four harmonized emission scenarios from different integrated assessment model (IAM) modeling groups (van Vuuren et al., 2011a; Meinshausen et al., 2011; van Vuuren et al., 2011b; Thomson et al., 2011; Masui et al., 2011; Riahi et al., 2011).

In a second step following the selection of concentration / emissions scenarios, five different socio-economic storylines were developed, the Shared Socio-Economic Pathways (SSPs: Nakicenovic et al., 2014), allowing mitigation and impact re-

searchers to combine low and high emission futures with different assumptions about socio-economic development in terms of population, Gross Domestic Product (GDP) and further indicators (van Vuuren et al., 2014). This is an advancement over the SRES scenarios, as in the SRES scenarios, an emissions future was often assumed to be in line with a single socio-economic development only. The exception during the SRES scenario process was the A1 scenario family that was during the plenary adoption process split out into three sub-scenarios, A1FI, A1T and A1B, indicating the importance of socio-economic assump-

tions below the high-level "high growth" storyline of the A1 family and their respective effect on emissions. The new SSP socio-economic storylines were modeled by several independent groups to quantify them in terms of GDP (Leimbach et al., 2017; Dellink et al., 2017; Crespo Cuaresma, 2017), population (KC and Lutz, 2017) and urbanization (Jiang and O'Neill, 2017) development on a country or detailed regional level. These scenario quantifications are called the SSP basic elements.

As a further step in this process, several research groups used IAMs to simulate combinations of the Shared Socio-Economic

Pathways with Representative Concentration Pathways forcing targets (Riahi et al., 2017; Kriegler et al., 2017; Fujimori et al., 2017; Calvin et al., 2017; Fricko et al., 2017; van Vuuren et al., 2017; Rogelj et al., 2018).

While the IAMs internally use between 11 and 26 regions, the published data are limited to a set of macro regions, namely the RC5 regions (RC: Region Categorization, see Edenhofer et al., 2014, AppendixII.2.2) for the RCPs, and the RC5.2 regions (IIASA, 2016) for the SSPs. The reasons for this limitation are manifold: decisions of the intercomparison protocols to allow a

wide participation of modeling groups lead to a neglect of some regional detail; but more fundamentally, the quality of calibration and input data for the global modeling exercises that produced the SSP GDP and population projections degrades on finer scales and hence limits the projection models. Furthermore, so far there are no official and comprehensive emissions inventories for most countries that are categorized as Non-Annex I countries, as their reporting requirements under the UNFCCC have been very limited compared to those categorized as Annex I / industrialized countries.

This limitation of country-level detail can severely hamper a number of studies: climate impact assessments, quantification of equity principles for effort-sharing of mitigation, or the assessment of pledges of countries against benchmark reference and mitigation scenarios. The required long-term country-level scenarios are only available based on the now over a decade old SRES scenarios (van Vuuren et al., 2007; Höhne et al., 2010).

Sector and gas resolution is limited as well. While the RCP scenarios have detailed sectoral data for some gases (e.g.

$CH_4$), the resolution of $CO_2$ is limited to separating land use emissions from the fossil fuel and industrial emissions in the publicly available database. The shared SSPv2 IAM outputs only resolve between land use and fossil / industrial emissions and hence also that coarse disaggregation is harmonized towards common historical emission levels. The RCPs resolve individual flourinated gases, while the SSPv2 database only provides data for aggregated fluorinated gases.



Recently, the scenarios for the Coupled Model Intercomparison Project phase 6 (CMIP6: World Climate Research Programme, 2019) have been released that are also based on the RCP forcing levels and SSP storylines. They provide socio-economic and emissions data on a more detailed regional, gas, and categorical level. Even though data exist that are downscaled to a grid - with an intermediate step of downscaling to country-level (Gidden et al., 2019) - there are no country-resolved data available. Transformation of gridded data to country-level is problematic for small countries unless the grid is very fine. Furthermore, only a few combinations of RCP forcings with SSP storylines each from a single model only are included in the SSP CMIP6 database (IIASA, 2018; Gidden et al., 2019; Feng et al., 2020).

To fill this gap and provide country-level data for all RCP SSP combinations and IAMs we downscale the RCPs and SSPv2 emissions scenarios to country-level using the SSP basic elements socio-economic country-level data. To downscale the RCPs we use per country GDP results from all three groups providing in SSP basic elementsGDP scenarios. Our data thus enable a comparison of results between SSP basic elementsmodeling groups and IAMs and provide ranges for future country emissions under different SSP storylines and RCP forcing targets instead of the seeming certainty given by the single model used in the SSP CMIP6 data.

Historical emissions data are taken from the PRIMAP-hist v2.1 source which provides data for all countries and Kyoto greenhouse gases (Kyoto GHGs) (Gütschow et al., 2016, 2019) based on official UNFCCC data complemented by third party data to fill the reporting gaps for non-Annex I countries and years before 1990. Historical socio-economic data are taken from Gütschow (2019), which is based on UN population data (UN DESA / Population Division, 2019) and the Maddison Project database (Bolt et al., 2018a, b) as well as other sources to fill missing values. Our downscaling methodology is based on existing approaches, which we extend and improve to enable the use on scenarios with negative emissions.

The paper is structured as follows: we begin with the review of existing downscaling methodologies and the introduction of our methodology in Section 2. In the following Section 3 we describe the data sources this work is based on and how they are processed. Section 4 presents a detailed step by step description of our downscaling approach. Results are presented in Section 5 followed by a discussion of limitations (Section 6) and conclusions (Section 7). The availability of the resulting datasets is described in Section 8. The appendix gives details on the data sources for the historical socio-economic data, data coverage of the different scenarios, and additional methodological details.

## 2 Methods

### 2.1 Notation

In the following we consistently assume data are given for a region $\mathbb{R}$ which we describe as a set of countries $\mathcal{C} \in \mathbb{R}$. We denote this by subscript identifiers. A regional emissions pathway is denoted by $\mathrm{E}_{\mathbb{R}}$, an emissions pathway for a specific country $\mathcal{C}$ is denoted by $\mathrm{E}_{\mathcal{C}}$. Emissions for a specific year $y$ are denoted by $\mathrm{E}_{\mathbb{R}}(y)$ or $\mathrm{E}_{\mathcal{C}}(y)$ respectively. We denote emission intensity by EI, GDP by GDP, and population by POP in a similar way. For calculations with full pathways we assume that the same operation is applied on datapoints for all years individually, i.e. $\mathrm{EI} \cdot \mathrm{GDP}$ denotes the multiplication of emission intensity by GDP for each year. $\mathrm{EI} \cdot \mathrm{GDP}(y)$ denotes the multiplication of the whole emission intensity time series by the GDP of year $y$. Emissions





and emission intensities are defined for several variables (gases, pollutants), but as we are only working on one gas at a time, we do not introduce another subscript index for these variables for the sake of a simpler notation. The method could as well be used to downscale the world to regional level or country emissions to state level. We only consider downscaling from larger to

smaller economic or political regions, e.g. from region level to country-level and do not consider spatial downscaling of data from coarser to finer grids. However, if, e.g. GDP data are given on a finer grid than emissions data, the method described here could also be applied.

We denote the RCP scenarios and forcing levels by RCP and the downscaled RCP scenarios by RCPd. The SSP basic elements are abbreviated by SSPbe. By SSPv2 we denote the SSP IAM scenarios version 2 and by SSPv2d the downscaled

SSPv2 scenarios. When using just SSP we refer to the SSP storylines, e.g. RCP SSP refers to the combination of RCP forcing levels with SSP storylines.

## 2.2   Existing downscaling methods

Several methods to downscale emissions data are found in the literature. Which methods can be used depends on available data and the choice between a simple and transparent method versus a more realistic but also more complex approach. Common to

all methods is the need for an auxiliary dataset called the downscaling key. Data from the downscaling key are used directly or as the basis for a model to split the regional data to country-level. It could be data for the same variable from a different source, or for a different variable with some known or assumed correlation to the variable that is to be downscaled. The data can either cover the same period of time, historical years only, or even a single year only. In van Vuuren et al. (2007) three groups of methods were identified which differ in the their use of the downscaling key. Here, we specifically consider cases,

where country-resolved emissions data are available up to a certain year, but future projections are only available for larger regions as this is the situation given by the combination of RCP scenarios with the SSP basic elements and the SSPv2 IAM runs.

**Linear downscaling**  This is the simplest method. The downscaling key is a dataset for the same variable as the to be down-scaled data, e.g., both $CO_2$ emissions. Historical emission data for one single year $y_0$ (or an averaged period) is used

to define shares $S_{\mathcal{C}}(y_0) = E_{\mathcal{C}}(y_0)/E_{\mathbb{R}}(y_0)$ for each country $\mathcal{C} \in \mathbb{R}$. These shares $S_{\mathcal{C}}(y_0)$ are used to distribute emissions from the regional pathway to individual countries: $E_{\mathcal{C}} = S_{\mathcal{C}}(y_0)E_{\mathbb{R}}$. The relative emissions of countries within a region are thus fixed at the historical level for the whole resulting scenario. This approach was used by the MATCH group (Höhne et al., 2010) to downscale SRES scenarios from region to country-level.

While this approach is very transparent and straight forward it has the downside that it can not model differing devel-

opments within a region. All countries in a region will have the same emission growth rates defined by the regional pathway. The method is likely to overestimate future emissions of relatively developed countries compared to those of developing countries with high economic growth within the same region. See also results Section 5.

**External input based downscaling**  In this method a country-resolved key pathway $K_{\mathcal{C}}$ for some variable is available. The shares $S_{\mathcal{C}} = K_{\mathcal{C}}/K_{\mathbb{R}}$ defined by this pathway are used to downscale the regional pathway: $E_{\mathcal{C}} = S_{\mathcal{C}}E_{\mathbb{R}}$.



This method can take different developments within the region into account, but only to the extent the downscaling key data K does itself. The intra-regional differentiation will be that of the existing key source, only scaled with the ratio of the regional scenario pathway to the regional key pathway. If the key data are for a different variable than the regional data to be downscaled, a systematic error is introduced if the two variables are not linearly correlated. If the correlation is known, this might be compensated, but in general this will not be the case. We use this method to downscale the SSPv2

socio-economic data and the PIK SSPbe data from region to country-level.

**Convergence downscaling** Convergence downscaling uses the assumption that a given variable converges among countries within a given region. The convergence assumption only makes sense for variables which are independent of the size of a country, e.g. emission intensity (emissions per unit of GDP) and GDP per capita but not absolute emissions or GDP. This method needs historical information for the target variable (e.g. emissions) and in case the target variable is

not independent of country size an auxiliary variable that can be used to create a convergence variable which does not depend on country size (e.g. GDP to create emission intensity). Furthermore, regional and country time-series for the auxiliary variable are needed for the full downscaling period.

The downscaling process begins with the creation of a temporary pathway of the convergence variable for all countries, starting from the historical values for each country and ending at a common value obtained from the given regional

pathway. Thus, all countries converge to the regional value in the convergence year. The convergence year can be set depending on the scenario storyline and governs if full or partial convergence is achieved within the scenario time frame. To accomplish partial convergence, the convergence year is set after the end of the scenario time frame and thus some form of extrapolation of the regional data is needed. In case we used an auxiliary variable we need to multiply the obtained pathways by the pathways of the auxiliary variable to obtain the temporary pathways for the downscaling

variable. The obtained temporary pathways are scaled such that their sum matches the regional pathway prescribed by the scenario for every year individually.

Convergence downscaling was employed by van Vuuren et al. (2006, 2007) to downscale the SRES scenarios from region to country-level. As the SRES scenarios do not contain any socio-economic data on the country-level, an external source had to be used. Regional SRES population data was downscaled using external input based downscaling with country-

level UN population projections. The GDP and emissions time series were then created using convergence downscaling.

This method employs socio-economic scenarios as the drivers of the downscaling process and is therefore a promising candidate to downscale the RCP and SSPv2 scenarios using the SSP basic elements country-level data. Details are presented in Section 2.3.

     Figure 1 shows examples for the three methods described above. Which method is most appropriate depends on intended

use and available data. If only historical data are available, linear downscaling is often the only method that can be used to derive country-level future emissions from regional emissions projections. Convergence downscaling is a good option, if the variable that should be downscaled can be expressed relative to some known variable to make it comparable between different



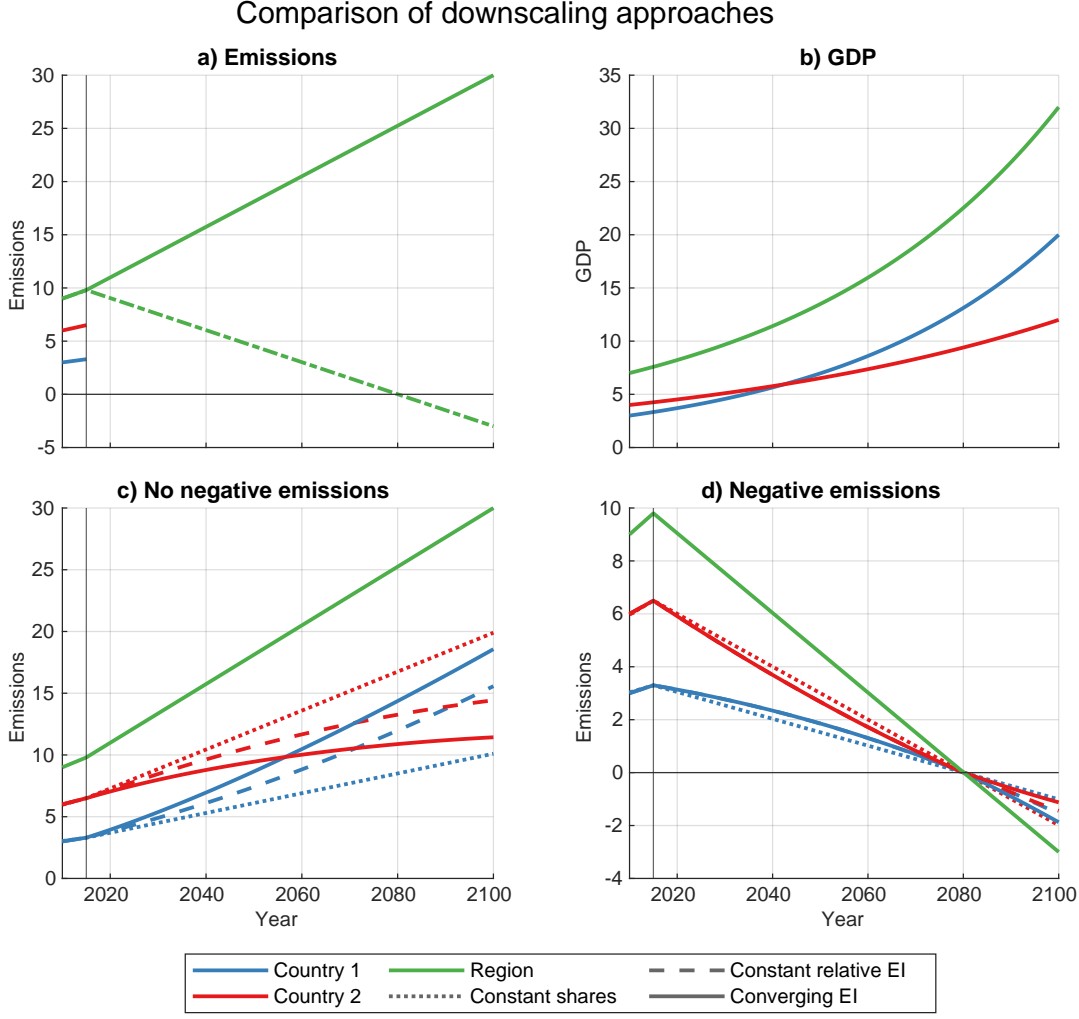

**Figure 1.** Example results for different downscaling approaches. For the sake of simplicity a two country region is assumed. The constant relative emission intensity downscaling is a variation of the external input based downscaling where we use GDP as external input and the assumption of constant relative emissions intensities to create emissions pathways based on the GDP data (Panel b). Regional emission data are given for the whole time period while for the countries only historical data are available (Panel a). Panel c shows downscaling for increasing emissions. It is clearly visible that the constant shares downscaling does not account for the GDP development, while the convergence downscaling leads to the highest emissions of country 2 because it not only considers GDP growth, but also converging emissions intensities between the two countries. Panel d shows downscaling for a transition to negative emissions. For convergence downscaling the convergence is set to the year directly before the transition to negative emissions. The rapid reductions and early convergence lead to similar pathways for all methods before the transition to negative emissions. After the transition the effect of considering GDP is visible. The convergence year for convergence downscaling is 2150 in this example.



countries, which is a prerequisite for the convergence concept to be meaningful. If emission data are available from a different source the external input method is a good option.

For our task we use a slightly modified version of the convergence downscaling which can handle negative emissions and uses the GDP data provided by the SSP basic elements and the IPAT equation to downscale the emissions of the greenhouse gases included in the RCP and SSPv2 scenarios. Our method is very similar to the convergence downscaling employed in van Vuuren et al. (2006, 2007) (see Section 2.3).

### 2.3  IPAT convergence downscaling

In this section we present the details of our modified version of the IPAT based convergence downscaling introduced in van Vuuren et al. (2006, 2007).

Similar to the original approach the basis for the downscaling of emissions is given by the IPAT equation (Ehrlich and Holdren, 1971; Chertow, 2000):

$$I = P \cdot A \cdot T. \tag{1}$$

The idea behind the equation is to decompose an environmental impact $I$ into its drivers. The IPAT equation assumes $I$ is linear in all three drivers: the population size $P$, the affluence $A$ as a measure of consumption of goods per capita, and a technology factor $T$ which governs the environmental impact per unit of consumed goods. In our case the environmental impacts to be described are greenhouse gas emissions. As we work on an economy wide level the affluence is described by GDP per capita and the emission intensity of the GDP plays the role of the technology factor. The driver behind emissions growth is total GDP

(as a measure of consumption and production) not the size of the population.

Our IPAT equation variant thus becomes

$$\mathrm{E}_\mathcal{C} = \mathrm{POP}_\mathcal{C} \frac{\mathrm{GDP}_\mathcal{C}}{\mathrm{POP}_\mathcal{C}} \frac{\mathrm{E}_\mathcal{C}}{\mathrm{GDP}_\mathcal{C}} = \mathrm{GDP}_\mathcal{C} \cdot \mathrm{EI}_\mathcal{C}. \tag{2}$$

where $\mathrm{EI}_\mathcal{C} = \mathrm{E}_\mathcal{C}/\mathrm{GDP}_\mathcal{C}$ is the emission intensity of country $\mathcal{C}$, the emissions per unit of GDP. The downscaling is carried out individually for each gas $g$ (index omitted).

Figure 2 gives an overview over the steps of the downscaling process, which will be described in detail in the following sections.

#### 2.3.1  Convergence and target emission intensity

The year of convergence for the emission intensity within a region has to be chosen according to the SSP scenario storyline. We assign relatively early convergence years (e.g. 2150) to scenarios with high economic integration, while scenarios with a

regionalization storyline only justify partial convergence within the scenario time frame. In case convergence is achieved during the scenario time frame, all countries within a region converge to the regional emission intensity prescribed by the emission scenario. In case of partial convergence we need to assume a regional emission intensity in a year after the end of the scenario. In van Vuuren et al. (2006, 2007) this was created using an exponential pathway with the average growth rate of the last years

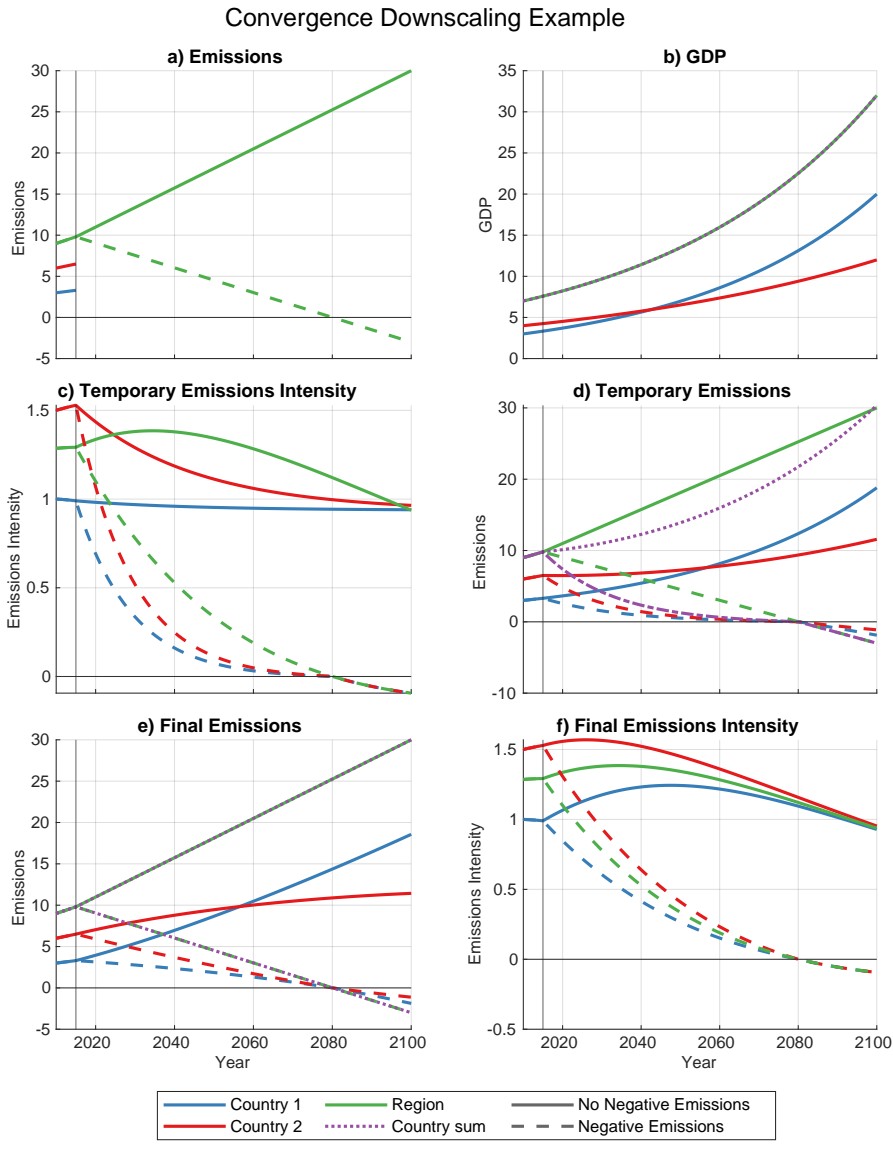

**Figure 2.** Steps of convergence downscaling of regional emissions data using the IPAT equation and country GDP data for a two country region for positive and negative regional emissions. Regional emission data are given for the whole time period while for the countries only historical data are available (Panel a). GDP data are given for both countries and the region for the whole time period (Panel b). In the first step temporary emission intensity pathways for the countries are calculated using exponential convergence from historical values (2015). In case of completely positive regional pathways, emissions intensities converge to the regional value in a given convergence year (2150, Panel c). In case of negative emissions convergence to the regional emission intensity is in the last year before the transition to negative emissions. After that year regional emissions intensities are used. Multiplication with the given GDP time series creates temporary emissions time series (Panel d). These do not sum up to the regional values (see Panel d) and have to be scaled to the regional value (results in Panel e). This also changes the emissions intensities (Panel f).





of the scenario. We judge exponential extrapolations to be very uncertain for long periods especially when the variable to be
extrapolated increases over time (as would be the case for e.g. $CO_2/GDP$ for e.g. the RCP 2.6 emissions scenario with SSP 4
basic elements GDP, Asia region). We therefore use the emission intensity of the last scenario year as target emission intensity
if the convergence year is after the end of the scenario time frame. For time series with a transition to negative emissions
we have to adjust the convergence year to avoid numerical instabilities and early (before regional total) transition to negative
emissions for countries with emissions intensities below the regional average. While this would make sense for countries which
base their low emissions intensity on a large share of renewable energy it is not realistic for developing countries with very
low emission intensities stemming from a low level of industrialization. We adjust the convergence year to be just before the
regional transition to negative emissions.

### 2.3.2   Construction of the temporary emission intensity pathways

To generate the per country temporary emission intensity pathways, we need a method to interpolate between the initial emis-
sion intensity given by historical data and the target emission intensity in the convergence year given by the regional scenario.
The methodology described in the following paragraphs is also presented graphically in Figure 2.

Our method is based on the original approach by van Vuuren et al. (2007). The emission intensity pathway of a country is
created using an exponential function that is defined by the initial emission intensity in the harmonization year and the regional
emission intensity in the convergence year. The idea is that change in emission intensity is proportional to the difference of
each country's emission intensity to the regional average.

The exponential convergence is modeled by the function

$$\widehat{\mathrm{EI}}_{\mathcal{C}}(y) = a_{\mathcal{C}} e^{\gamma y} + b_{\mathcal{C}}, \quad \text{for } y_h < y < y_c, \tag{3}$$

where $y_h$ denotes the year of harmonization with historical data and $y_c$ denotes the convergence year. The decay factor $\gamma$ is
defined as

$$\gamma = \frac{\ln(d)}{y_c - y_h}, \tag{4}$$

such that the exponential function reduces the difference between country and regional emissions intensities

$$\mathrm{EI}_{\mathrm{diff}}(y_h) = \mathrm{EI}_{\mathcal{C}}(y_h) - \mathrm{EI}_{\mathbb{R}}(y_h) \tag{5}$$

in the harmonization year $y_h$ to $\mathrm{EI}_{\mathrm{diff}}(y_c) = d\mathrm{EI}_{\mathrm{diff}}(y_h)$ in the convergence year $y_c$, where we chose $d = 0.01$ to have almost
complete convergence. Smaller values would lead to rapid partial convergence in the first years with only small changes in the
later years. The country specific constants $a_{\mathcal{C}}$ and $b_{\mathcal{C}}$ are defined via

$$a_{\mathcal{C}} = \frac{\mathrm{EI}_{\mathbb{R}}(y_c) - \mathrm{EI}_{\mathcal{C}}(y_h)}{e^{\gamma y_c} - e^{\gamma y_h}} \tag{6}$$

and

$$b_{\mathcal{C}} = \frac{\mathrm{EI}_{\mathbb{R}}(y_c) - d\mathrm{EI}_{\mathcal{C}}(y_h)}{1 - d}. \tag{7}$$





In case of convergence before the end of the scenario time span we continue all country time series with the regional emission intensity:

$$\widehat{\mathrm{EI}}_\mathcal{C}(y) = \mathrm{EI}_\mathbb{R}(y), \quad \text{for } y \geq y_c. \tag{8}$$

For the part of the scenario also covered by historical data we use the historical emission intensity:

$$\widehat{\mathrm{EI}}_\mathcal{C}(y) = \mathrm{E}_\mathcal{C}(y)/\mathrm{GDP}_\mathcal{C}(y), \quad \text{for } y \leq y_h. \tag{9}$$

As an alternative to exponential convergence we also studied linear convergence of emission intensities. However we were not able to produce sensible results as the scaling step (Section 2.3.3) exhibited numerical instabilities.

The result of this step is a set of temporary emission intensity pathways $\widehat{\mathrm{EI}}_\mathcal{C}$ for every country $\mathcal{C} \in \mathbb{R}$.

### 2.3.3 Emission pathways and scaling

Using the IPAT equation, we generate a preliminary emissions pathway $\widehat{\mathrm{E}}_\mathcal{C}$ for every country $\mathcal{C}$:

$$\widehat{\mathrm{E}}_\mathcal{C} = \mathrm{GDP}_\mathcal{C}\widehat{\mathrm{EI}}_\mathcal{C}. \tag{10}$$

Those pathways are summed up to a preliminary pathway for the region $\mathbb{R}$:

$$\widehat{\mathrm{E}}_\mathbb{R} = \sum_{\mathcal{C} \in \mathbb{R}} \widehat{\mathrm{E}}_\mathcal{C}. \tag{11}$$

In general this pathway will differ from the regional pathway prescribed by the scenario. We create a scaling pathway

$$S_\mathbb{R} = \mathrm{E}_\mathbb{R}/\widehat{\mathrm{E}}_\mathbb{R}. \tag{12}$$

The final country pathway is defined via

$$\mathrm{E}_\mathcal{C} = \widehat{\mathrm{E}}_\mathcal{C}S_\mathbb{R}. \tag{13}$$

This method does not work in case of negative emissions which are common for $CO_2$ pathways in low emissions scenarios like RCP 2.6 or the new $1.9W/m^2$ scenarios (Rogelj et al., 2018) where technologies like bioenergy with carbon capture and storage (BECCS) are assumed to remove large quantities of $CO_2$ from the atmosphere. So the temporary $CO_2$-emission pathways (see Equation 10) of all countries in a region with negative emissions contain a transition to negative emissions and so does the regional sum pathway (Equation 11). Similarly, the regional pathway will be near zero for a few years before and after it's transition to negative emissions. Therefore, the calculation of the scaling pathway (Equation 12) is numerically unstable. As the country pathways are not necessarily near zero where their sum is zero, some adjusted country pathways will exhibit positive peaks in emissions, while others will contain negative emission peaks, still summing to the correct regional value. These peaks are several years wide and can not be removed by interpolation without major changes to the country pathways.

We have investigated several different options to circumvent this problem including dynamical downscaling algorithms which downscale data year by year and can use alternate algorithms when regional emission intensity is near zero. However, fine



tuning the parameters to deal with the transition to negative emissions for several scenarios proved to be very complicated while the results were often very similar to the very simple solution of moving the convergence year to before the transition to negative emissions. After that year all countries follow the same emission intensity pathway. The steep reduction in emissions and emission intensity does not leave much freedom for the downscaling (see also Figure 1). All countries have to rapidly reduce emissions to meet the prescribed regional pathway. Furthermore, as described in Section 2.3.1 there are conceptual problems with convergence years set to later than the transition to negative emissions. We therefore use the simple but transparent approach of early convergence. The calculation itself is not changed but $y_c$ is adjusted to be the last year before the transition to negative emissions. This is done on a per gas level. Only $CO_2$ pathways have negative emissions, consequently only $y_c$ of $CO_2$ is adjusted. The downside of this approach is that it impacts the assumptions of convergence and eliminates the possibility to define different convergence speeds for different socio-economical storylines. Figure 2 gives an overview over the steps of the downscaling process.

## 3  Input data and preprocessing

This section provides an overview over the input data. It covers the RCP and SSP scenarios (Section 3.1) and their implementation including the choice of scenarios for international shipping and aviation (Section 3.1.3), the region definitions used in the models (Section 3.3), the countries covered by the datasets (Section 3.3), and the covered sectors and gases (Section 3.4). Furthermore, the historical data used to downscale the RCP and SSP scenarios are introduced in Section 3.2.

### 3.1  Scenario description

Two datasets are produced based on two sets of scenarios: RCPd, based on the RCP scenarios (van Vuuren et al., 2011a), which are downscaled using the SSP basic elements and SSPv2d, based on the SSPv2 IAM implementations of the SSP scenarios, which come with consistent socio-economic data that are used for downscaling (Riahi et al., 2017; Rogelj et al., 2018).

Not all combinations of RCP GHG forcing scenarios and SSP storylines are meaningful, as some SSP storylines imply, e.g., emissions that lead to forcing levels below RCP 8.5 while other SSPs imply high unmitigated emissions, which are unrealistic to be mitigated to the lowest RCP forcing levels without substantially changing the socio-economic storyline. For the SSPv2 scenarios the possible combinations were determined by the IAMs: the SSP-specific baseline scenarios define the maximal forcing level for each SSP while the minimal level was found implicitly because the forcing level of low RCPs could not be attained for all SSPs.

### 3.1.1  SSPv2 IAM runs (SSPv2d)

During the integrated assessment model (IAM) implementations of RCP SSP combinations (SSPv2) it was found that some combinations can not be implemented. Figure 8 in Riahi et al. (2017) illustrates the carbon prices needed to reach a certain mitigation level under a given SSP. The figure also shows that the RCP 8.5 forcing is only reached for SSP 5. All other SSPs have baseline emissions leading to a lower climate forcing. For SSP 1, RCP 6 is the baseline, for SSPs 2-4 the baseline





forcings are between RCP 6 and RCP 8.5. Under SSP 3, the low emissions scenario RCP 2.6 can not be attained and under
SSP 5 one model was unable to attain sufficiently low emissions. The IAM implementations of SSP scenarios use an additional
intermediate forcing level of $3.4W/m^2$ (Riahi et al., 2017), which can be reached under all SSP storylines. Additionally, SSPv2
contain a new strong mitigation pathway reaching a forcing level of only $1.9W/m^2$ (Rogelj et al., 2018). This forcing level is
attained for SSPs 1, 2, and 5. Only a single model could attain a forcing of $1.9W/m^2$ under SSP 4 and no model under SSP 3
(see Rogelj et al., 2018, Fig. 5).

   We downscale all SSPv2 runs, both marker and other. An overview is shown in Table 1. However, not all scenarios have
been implemented by all modeling groups (see Appendix A1). For each SSP a different IAM provided the respective illustrative
marker scenario. Namely those were for SSP 1: IMAGE (van Vuuren et al., 2017); SSP 2: MESSAGE (Fricko et al., 2017);
SSP 3: AIM-CGE (Fujimori et al., 2017); SSP 4: GCAM4 (Calvin et al., 2017); SSP 5: REMIND-MAGPIE (Kriegler et al.,
2017).

**Table 1.** Forcing levels attained by SSPv2 studies. See Riahi et al. (2017), Calvin et al. (2017), and Rogelj et al. (2018). For SSP 1, RCP 6 is
the baseline forcing for all models except WITCH, which has a slightly higher baseline such that RCP 6 is a mitigation scenario. "x" means
that the forcing level could be attained by all models that implemented it, and "(x)" that it could be attained by at least one model but not in
the marker implementation. All forcings in $W/m^2$.

|            | SSP 1     | SSP 2     | SSP 3     | SSP 4  | SSP 5  |
|------------|-----------|-----------|-----------|--------|--------|
| Baseline   | 5.8       | 6.5 – 7.3 | 6.7 – 8.0 | 6.4    | 8.5    |
| RCP 8.5    |           |           |           |        | x=BL   |
| RCP 6      | (x)=BL    | x         | x         | x      | x      |
| RCP 4.5    | x         | x         | x         | x      | x      |
| $3.4W/m^2$ | x         | x         | x         | x      | x      |
| RCP 2.6    | x         | x         |           | x      | (x)    |
| $1.9W/m^2$ | x         | x         |           | (x)    | x      |
| Marker     | IMAGE     | MESSAGE   | AIM-CGE   | GCAM4  | REMIND |

### 3.1.2    RCP and SSP basic elements (RCPd)

To select sensible combinations of RCP scenarios and SSP basic elements scenarios we use the SSPv2 IAM runs as a basis.
The combination of RCP 8.5 with SSP 1 is excluded because no model reached emissions significantly above RCP 6 levels and
the SSP 1 storyline of a rapid sustainable development is not compatible with RCP 8.5 emission levels. The baseline forcings
of SSPs 2-4 do not reach $8.5W/m^2$, however, forcings are significantly above $6W/m^2$ for SSP 2 ($6.5W/m^2 – 7.3W/m^2$) and
SSP 3 ($6.7W/m^2 – 8.0W/m^2$) (Riahi et al., 2017). Thus we include the combination of RCP 8.5 with SSPs 2 and 3. SSP 4
models a very unequal socio-economic development with low reference emissions as only a small part of the world has high



consumption levels and cheap mitigation options as investment in new technologies is high. The baseline forcing of $6.4 W/m^2$ (Calvin et al., 2017) is above RCP 6 but significantly below RCP 8.5. We thus exclude the combination of SSP 4 with RCP 8.5.

In the IAM studies it was also found that the SSP 3 storyline does not allow for sufficient mitigation to reach RCP 2.6 forcing levels (Riahi et al., 2017). Consequently, we exclude this combination. The combination of SSP 5 with RCP 2.6 is included as most models used for SSPv2 can attain the necessary forcing levels. All combinations considered are shown in Table 2.

**Table 2.** Combination of RCP scenarios with SSP basic elements country results considered in this study.

|           | SSP 1 | SSP 2 | SSP 3 | SSP 4 | SSP 5 |
|-----------|-------|-------|-------|-------|-------|
| RCP 8.5   |       | x     | x     |       | x     |
| RCP 6     | x     | x     | x     | x     | x     |
| RCP 4.5   | x     | x     | x     | x     | x     |
| RCP 2.6   | x     | x     |       | x     | x     |


Figure 3 gives an overview over the RCP and SSPv2 scenarios. Figures for individual gases are available in the SI, Section 2.1.

### 3.1.3 Emissions from international shipping and aviation

Emissions from international shipping and aviation (bunker fuels) are not attributed to individual countries under the UNFCCC.
Therefore they need special consideration in the downscaling process.

**RCPd** Emissions from international shipping and aviation are included in the RCP scenario emissions. For $CO_2$ and $N_2O$ (marine only) however, they are not provided as individual emission time series, but included into the regional emissions. As growth rates of emissions from aviation and shipping likely differ from growth rates of general fossil $CO_2$ emissions, the inclusion changes the growth rates of the regional emissions pathways. As there are no readily available consistent
$CO_2$ pathways for international shipping and aviation for the original RCP scenarios, they have to be either generated or taken from other scenarios. The RCPs provide data for several gases and pollutants for aviation and shipping. It suggests itself to try to calculate $CO_2$ emissions consistent with the RCP emissions from other gases using correlations between $CO_2$ and the other gases obtained from scenarios which cover all gases. However, using the shipping and aviation time series from Owen et al. (2010) and QUANTIFY (2010) to compute the correlations, no consistent $CO_2$ pathways could
be generated as results based on different gases were not consistent. We therefore have to use external scenarios. We use the CMIP6 emissions scenarios from Gidden et al. (2019) which are based on the RCP forcing and SSP storylines and are consistent with the RCPs on a basis of the RCP forcing targets but not the pathways to reach these targets. See Table 3 for our choice of CMIP6 bunkers scenarios for the RCPs.

The CMIP6 scenarios contain emissions for international shipping for $CO_2$ and $CH_4$ as well as aviation emissions for
$CO_2$. Unfortunately $N_2O$ emissions are only given as a national total. We thus compute a $N_2O$ over $CO_2$ factor from



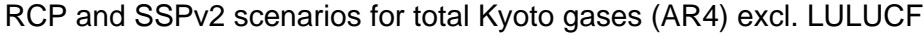

**Figure 3.** RCP and SSPv2 scenarios for total Kyoto GHG emissions (AR4 GWPs) excluding LULUCF. Scenarios are not harmonized to historical data. Historical data shown are from PRIMAP-hist with bunker fuel $CO_2$ emissions added from CDIAC data (Boden et al., 2017; Andres et al., 1999; Marland and Rotty, 1984).





historical data (2007 – 2012 average from Smith et al., 2014) and construct $N_2O$ emissions only contribute roughly 1% of total bunkers emissions. scenarios from the $CO_2$ scenarios assuming this factor is constant over time. *NtO* To downscale total aviation emissions to domestic and international aviation we use the shares from the historical CMIP6 emission data (Hoesly et al., 2018).

**SSPv2d** The SSPv2 scenarios as presented in the SSPDB (IIASA, 2018; Riahi et al., 2017) do not include explicit bunkers emissions. As for the RCPs we use the CMIP6 scenarios which also offer implementations of the new $1.9W/m^2$ and $3.4W/m^2$ forcing targets. We base the bunkers emissions on the forcing targets only and use the same emissions time series for all SSPs. The methods are generally the same as for the RCP scenarios with a few small adjustments. External scenarios are needed for all gases including $CH_4$ which has explicit data in the RCP scenarios but not in SSPv2. For 330 the SSP baseline scenarios we use the SSP 3 baseline implementation reaching $7.0W/m^2$ for all SSPs. The choice of scenarios is shown in Table 3.

In the scenarios from two of the models (IMAGE and AIM-CGE) there is a slight ($< 1GtCO_2eq$) discrepancy between global emissions and the sum of the regional emissions. The discrepancy is decreasing in time towards 2100. International bunkers are a good explanation for additional global emissions. However, the discrepancies are much smaller than any 335 bunkers estimate, especially in the future. We thus discard the global data and work with the regional data as for all other scenarios.

**Table 3.** Choice of SSP CMIP6 bunkers pathways to complement the RCPs and SSPv2 scenarios with $CO_2$, $N_2O$, and $CH_4$ emissions. $CH_4$ is only needed for the SSPv2 scenarios as data are available for the RCPs. Baseline emissions are between RCP 6 and RCP 8.5 for most SSPs.

| RCP | Aviation and shipping scenario |
|---|---|
| Baseline | SSP 3 7.0 BL AIM-CGE |
| RCP 8.5 | SSP 5 8.5 BL REMIND-MAGPIE |
| RCP 6 | SSP 4 6.0 GCAM4 |
| RCP 4.5 | SSP 2 4.5 MESSAGE-GLOBIOM |
| $3.4W/m^2$ | SSP 4 3.4 GCAM4 |
| RCP 2.6 | SSP 1 2.6 IMAGE |
| $1.9W/m^2$ | SSP 1 1.9 IMAGE |

The bunkers scenarios are shown in Figure 4.

To create scenarios excluding bunkers emissions we subtract the bunkers emissions from the regional pathways using the historical $CO_2$ bunkers emissions from CDIAC (Boden et al., 2017; Andres et al., 1999; Marland and Rotty, 1984) (2004 340 – 2014 average) to downscale the global aviation and shipping pathway to region level. This does not take into account the development of regional emissions and regional economies, but as it is unclear how the international bunkers emissions were



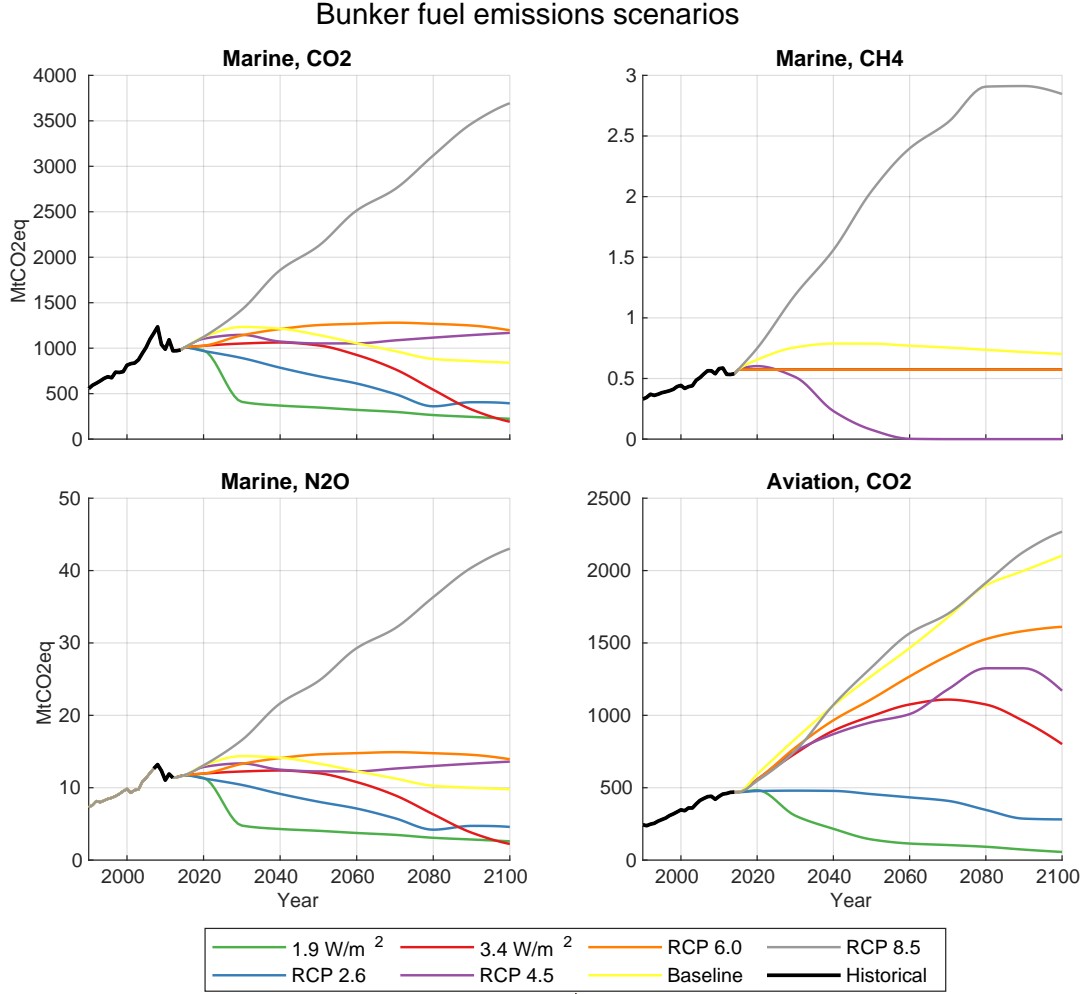

**Figure 4.** Scenarios for bunker fuels used to remove bunkers emissions from the RCP and SSPv2 emissions scenarios. Aviation emissions are only available for $CO_2$. Historical data are from CMIP6 (Hoesly et al., 2018) for $CO_2$, and $CH_4$. For $N_2O$ we use data from the International Maritime Organization and EDGAR v4.3.2 data (JRC and PBL, 2017; Janssens-Maenhout et al., 2019) for the time not covered by the IMO data.





calculated and assigned to regions when creating the RCP data, a more sophisticated method would not necessarily lead to better results.

Bunkers emissions also depend on the socio-economic storyline, not only the emissions scenario, so selecting the bunkers pathways solely based on the RCP forcing levels and not based on the SSP storylines, which govern, e.g. trade patterns, is a simplification. There are two reasons for this: firstly, for gases that are not well mixing (i.e. all except $CO_2$ and $N_2O$, but of these we only use $CH_4$ here) bunkers data are already given for the RCPs; secondly, there are no bunkers scenarios available for all different RCP SSP combinations, so basing the selection of bunkers scenarios on both RCP and SSP would require several assumptions.

## 3.2 Historical data

Our aim is to create a set of scenarios that is directly usable for climate policy research and analysis. It is important that the country specific pathways are in line with historical data for both emissions and socio-economic variables. We do not use the historical data provided with the RCP and SSP scenarios as we want to use latest historical compilation datasets (Gütschow et al., 2019, 2016).

### 3.2.1 Historical emissions data

We use the PRIMAP-hist (v2.1) historical emissions time series (Gütschow et al., 2016, 2019). It combines multiple data sources into one comprehensive dataset covering all Kyoto GHGs, all sectors, all countries, and all years from 1850 to 2017. Emissions data for some gases and sectors is interpolated for the last years. The highest priority during the combination of time-series from different sources is given to data which has been reported to the UNFCCC by countries. The dataset can be viewed on Paris Reality Check (PRIMAP, 2020) and is openly accessible (Gütschow et al., 2019).

### 3.2.2 Historical socio-economic data

We use the PRIMAP-hist historical socio-economic time series (Gütschow, 2019). It is constructed using the same methods as the PRIMAP-hist emissions time series.

For population data we use the UN population prospects (UN DESA / Population Division, 2019), and fill gaps and missing countries from the World Bank's World Development Indicators (WDI) database (The World Bank, 2019b, a). HYDE 3.2 data are used for extrapolation into the past until 1850 (Klein Goldewijk et al., 2017; Klein Goldewijk, 2017).

GDP data are based on purchasing power parity (PPP) adjusted data from the Penn World Table (Feenstra et al., 2015, 2019). Missing data are filled using the 2018 Maddison Project Database (Bolt et al., 2018b, a) and WDI. Finally we fill missing historical data from a processed version of the older Maddison Project data (Geiger, 2018; Geiger and Frieler, 2017; Bolt and van Zanden, 2014; Maddison Project, 2013). See also (Gütschow, 2019).





The choice of PPP adjusted GDP has two reasons: firstly, for compatibility reasons as the SSP data are given in PPP corrected form; secondly, PPP adjusted GDP is more comparable between countries than market exchange rate (MER) based GDP, which is important for the downscaling process as the process assumes convergence of emissions intensities.

### 3.3 Regions and country coverage

Here, we provide information on the regions used for the input data and the conditions under which countries are included in the input data and the final dataset. For a country to be available in the final time series it needs to be included in the SSP basic elements GDP time series, the historical data for both emissions and GDP and for the SSPv2 scenarios it further needs to be included in the region definitions of the IAMs. Table 2 of the SI gives an overview over available countries in each scenario, the input data, and the final dataset.

#### 3.3.1 RCPd

The emissions pathways provided with the RCP scenarios divide the world into 5 regions (IIASA, 2009):

**ASIA** Asian countries

**LAM** Latin America

**MAF** Middle east and Africa

**OECD90** The OECD countries as of 1990 and some pacific island states

**REF** Reforming economies (former Soviet Union)

Each of these regions is downscaled individually and not influenced by values from other regions. Where countries are missing in the socio-economic pathways or the historical emissions data, they are ignored and the regional emissions are split among the available countries. The SSP basic elementsdo not cover all countries but, depending on the modeling group, leave out
some smaller countries. This excludes several small states such as most of the small island states. Those states are therefore excluded from the downscaled dataset. A list of those countries can be found in the SI in Section 1.3.7. Some countries do not have data for all variables and are included in the final datasets with the available variables.

The socio-economic scenarios provided by the SSPbe modeling groups contain population (KC and Lutz, 2017) and GDP (Dellink et al., 2017; Leimbach et al., 2017; Crespo Cuaresma, 2017) projections on a per-country or detailed per-region level.
Population data are only provided by the IIASA group, the other groups (OECD and PIK) use the IIASA population projections to build their GDP projections. GDP data are provided in purchasing power parity (PPP) corrected form in 2005 international dollars (GKD)[1]. The PIK data are provided on a level of 32 world regions. We downscale it to individual country-level using the underlying IIASA population data and the method introduced in Section 4.2. In the SI, Section 1.2 we present the exact region definitions and list of missing countries for each modeling group.

---

[1]Actually, data are provided in 2005USD, but for a PPP corrected GDP this equals 2005GKD.





### 3.3.2 SSPv2d


The SSPv2 IAM implementations provide both emissions and socio-economic data on the level of 5 world regions similar to the regions used in the RCPs. However, the exact region definitions in terms of included countries differ from model to model. A detailed list with region definitions is available from the SSP database (IIASA, 2016). We use the model dependent region definitions to downscale both socio-economic and emissions data. GDP data are provided in PPP corrected form in

2005GKD/USD. The socioeconmic data of the SSPv2 runs is based on the IIASA country population data (KC and Lutz, 2017) and the OECD GDP data (Dellink et al., 2017). Consequently, we use these datasets to downscale the IAM data to country-level using an external input based downscaling method (see Section 4.3). In the SI, Section 1.3 we present a list of missing countries for each model.

### 3.4 Sectors and gases

The sector and gas resolution of the historical time series is finer than the resolution of the scenarios for all sectors, gases, and countries. Thus, the resolution of the final dataset is determined by the resolution of the scenario data. In this section we are only considering emissions time series as population and GDP are given as national totals.

LULUCF emissions are subject to high annual fluctuations and their development very much depends on individual country's policies. Furthermore, the scenario data often has positive emissions for regions for which historical data show negative

emissions in the past years. In this case the past (negative) emission shares and emission intensity are no indicator for projected (positive) emissions. LULUCF downscaling thus needs several strong assumptions which we think users of the data should make knowingly instead of unknowingly using our assumptions. In conclusion we exclude LULUCF data from the downscaling as done in van Vuuren et al. (2006, 2007).

### 3.4.1 RCPd

The RCPs include information for the Kyoto GHGs ($CO_2$, $CH_4$, $N_2O$, and the fluorinated gases (f-gases)) as well as several other substances (CO, $SO_2$, $NH_3$, $NO_x$, Black Carbon (BC), Organic Carbon (OC), Volatile Organic Compounds (VOC), and Ozone Depleting Substances (ODS)). Here we focus on the Kyoto GHGs because of their special relevance to the UNFCCC negotiations and availability of historical data. Additional substances can be added if there is demand from the scientific community and where historical data are available. (for historical data see Hoesly et al., 2018; Meinshausen et al., 2017).

Fluorinated gases are treated as one gas at the moment. Historical data for fluorinated gases is available at a level of aggregate HFCs, aggregate PFCs, and $SF_6$ for all countries, but to be consistent with the SSPv2 scenarios which only provide aggregate data for fluorinated gases we do not use this more detailed data in the downscaling process. Some substances such as black carbon need other downscaling methods as they are often co-emitted with gases like $CO_2$. This correlation of emissions has to be taken into account in the downscaling.

The sectoral detail of the emissions data provided with the RCPs differs between the greenhouse gases. The data for the most important gas, $CO_2$, is only resolved into emissions from land use, land use change, and forestry (LULUCF) and emissions





from fossil fuels and industry. We employ the same sectoral resolution for the other Kyoto GHGs. $N_2O$ data are only available as national total. As LULUCF emissions only constitute a fraction of roughly 3% of global $N_2O$ emissions (in 2015, see Gütschow et al., 2018) we use the total emissions as a proxy for fossil fuel and industrial emissions.

### 435 3.4.2 SSPv2d

In principle the SSPv2 scenarios cover the same substances as the RCP scenarios. However, fluorinated gases are only available as a global warming potential weighted aggregate time series. Therefore, fluorinated gases (f-gases) are treated as one substance. While the global warming potential (GWP) used for the f-gases basket is not explicitly given, the data are consistent with a Kyoto GHGs basket[2] created using GWPs from the IPCC's Fourth Assessment Report (AR4). Therefore, we assume 440 that the f-gases basket has been calculated based on AR4 GWPs.

In terms of sectors the SSPv2 scenarios offer less detail than the RCPs: $CO_2$, $CH_4$, and $N_2O$ emissions are available for national total and a sector called "land use" independently. For $CO_2$, and for some scenarios also for $CH_4$, additional time series for emissions from fossil fuels and industry are provided. However, the employed definition of the "land use" sector differs from the definition in the IPCC categorizations. The high emission levels for $CH_4$ and $N_2O$ suggest that, rather than for land use 445 only, the time series cover emissions from the Agriculture, Forestry, and Land Use (AFOLU) sector. For $CO_2$ this is no practical problem as agricultural $CO_2$ emissions contribute less than 0.1% to total $CO_2$ emissions (Gütschow et al., 2019) and we use the "land use" sector as a proxy for LULUCF. However, for $N_2O$ and $CH_4$ this is not possible as agricultural contributions are substantial. We thus use national total emissions as a proxy for fossil fuel and industrial emissions, as LULUCF emissions for these gases account for only 3% ($N_2O$) and 4% ($CH_4$) of national total emissions (Gütschow et al., 2018). Emissions of 450 fluorinated gases are available as national total only, which suffices as they originate from industrial sources only.

## 4 Downscaling of RCP and SSPv2 scenarios

The following describes the generation of the downscaled RCP and downscaled SSPv2 scenarios step by step from the preparation of input data to the combination of historical and scenario data for the final time series.

**Data preparation** The RCP and SSP data are processed as described in Section 4.1. Historical data does not need preprocess-455 ing at this step.

**Downscaling of socio-economic data** Not all socio-economic data have country resolution. The PIK GDP data need downscaling to country-level (Section 4.2) and the SSPv2 socio-economic data as well (Section 4.3). After the downscaling, all socio-economic data are processed to match the country definitions of the historical emissions data.

**Generation of socio-economic scenarios** In this step GDP and population time series from all SSP scenarios are combined 460 with historical data (Section 4.4). The socio-economic part of the dataset is finalized with this step and is used as input for the emissions downscaling.

---

[2]The Kyoto GHG basket is the GWP weighted sum of $CO_2$, $CH_4$, $N_2O$, and the fluorinated gases (HFCs, PFCs, and $SF_6$).



**Downscaling of RCP and SSPv2 emissions data** RCP data are downscaled using the SSP basic elementscountry data (Section 4.5), while SSPv2 data are downscaled using the downscaled SSPv2 socio-economic data (Section 4.3). During the process the downscaling key is harmonized to historical data (Section 4.4).

**Generation of emissions scenarios** In the final step the downscaled RCP and downscaled SSPv2 emissions scenarios are combined with and harmonized to historical emissions data. (Section 4.6).

All operations are carried out independently per scenario, region and gas. The combination of historical and scenario data is carried out independently per scenario, country, and gas.

## 4.1   Preparation of RCP and SSP data

RCP and SSP data have to be preprocessed such that data are available for all sectors, gases, and years needed for the downscaling.

### 4.1.1   RCP data

The RCP data only offers values every 10 years. It is interpolated using MATLAB's pchip function to obtain yearly values needed for harmonization. The data does not resolve any categories for $N_2O$ and fluorinated gases. National total values are
copied to obtain values for emissions excluding land use for fluorinated gases and $N_2O$. For Methane higher level categories are aggregated from the lower level categories available in the RCP data. We build the HFCs, PFCs, aggregate f-gases, and Kyoto GHGs baskets for GWPs from the IPCC's Second Assessment Report (SAR) and Fourth Assessment Report (AR4)

Time series excluding bunkers emissions are created in accordance with Section 3.1.3.

### 4.1.2   SSP basic elements

The SSP basic elements country-level data are first summed to the country definitions used for the historical GDP and population data. GDP data are given in PPP corrected 2005USD and have to be converted to 2011GKD (see Appendix B1. for details). As IIASA and OECD data cover a slightly different set of countries we create a composite GDP source which uses the OECD GDP data as the basis and fills missing countries from the IIASA data. See Table 2 of the SI for details.

### 4.1.3   SSPv2 socio-economic data

The SSPv2 socio-economic scenarios are interpolated to obtain yearly values from time series with a temporal resolution of 10 years. No further processing is done at this point.

### 4.1.4   SSPv2 emissions data

The SSPv2 emissions data are interpolated to obtain yearly values. $N_2O$ and $CH_4$ emissions excluding LULUCF are obtained from national total emissions. Existing time series are discarded because they do not include agricultural emissions. Flourinated





gases are only available as a AR4-GWP weighted sum. To create a time series for SAR GWPs, regional conversion factors
from AR4 to SAR are calculated from EDGAR v4.2 data for individual gases using the years 2000 to 2012. As for $CH_4$ and
$N_2O$, a copy of national total f-gases emissions is used for national total excluding LULUCF. We build Kyoto GHGs baskets
for SAR and AR4 GWPs.

   Time series excluding bunkers emissions are created in accordance with Section 3.1.3.

## 4.2   Downscaling of PIK GDP data

The PIK GDP data are not available on a per country-level, but only for 32 regions. These regions are downscaled to country-
level using external input downscaling (see Section 2.2) with country shares from the OECD GDP data (complemented by
IIASA data for missing countries).

   The results of the GDP downscaling are in line with the GDP projections of the two other modeling groups. The pathways
show similar developments and the spread between different scenarios is similar to the other modeling groups. This is to be
expected as we use the country data from OECD and IIASA as external input to the downscaling. For results see Figures 6 and
7.

## 4.3   Downscaling of SSPv2 GDP and population data

The population and GDP data used as input to the SSPv2 IAM runs are based on the SSP basic elementsresults. IIASA
population data (KC and Lutz, 2017) and OECD GDP data (Dellink et al., 2017) are used. In theory the data used by the IAMs
should be identical to the country model data, however, different region specifications can introduce small changes in the data.
We thus do not take the country data directly but use it as the key in an external input based downscaling of the IAM data:
the regional GDP and population time series from the IAM scenarios are downscaled to country-level using shares from the
country model results. We use model specific region definitions for the downscaling (see Section 3.3.2 and SI, Section.

## 4.4   Harmonization

Harmonization of scenario data to historical data is used in several places throughout this study. Whenever IPAT based down-
scaling is used, the downscaling key is harmonized to historical data. This is necessary to ensure that the concept of converging
emissions intensities holds for the final scenarios, in which both socio-economic and emissions data are harmonized to and
combined with historical data. Downscaling with external or constant shares does not need this harmonization step as both
methods do not use socio-economic data. For the RCP and SSPv2 emissions downscaling this means that the resulting down-
scaled data are consistent with the harmonized GDP data, not with the raw SSPv2 GDP data. The effect of GDP harmonization
is shown in the SI in Section 2.3.

   We also create time-series where scenario data are harmonized to historical data for socio-economic and emissions data.
The harmonization techniques and parameters are similar for socio-economic and emissions data. The harmonization year is
always 2017. For the historical value we do not directly use the 2017 data, but calculate a value using a 13-year linear trend





(2005 – 2017) to weaken the influence of short term fluctuations in data. From this value ($E_{h,\mathrm{hist}}$) and the 2017 scenario value $E_{h,\mathrm{scen}} = E_{\mathrm{scen}}(2017)$ a harmonization factor is calculated: $f_h = E_{h,\mathrm{hist}}/E_{h,\mathrm{scen}}$. For socio-economic data we use this harmonization factor to harmonize the whole time series:

$$GDP_h(y) = f_h \cdot S(y). \tag{14}$$

This amounts to using the scenario growth rates to extend the historical time series. For GHG emissions data we phase out the harmonization factor $f_h(y)$ linearly until a convergence year $y_c = 2050$. Thus, $f_h(y_h) = f_h$ and $f_h(y_c) = 1$ and linear interpolation between these values gives:

$$E_h(y) = f_h(y)E(y) = \left( \frac{(1 - f_h)y + (f_h y_c - y_h)}{y_c - y_h} \right) E(y). \tag{15}$$

We phase out the harmonization factor to both keep the cumulative emissions of the scenario close to it's design and achieve a
smooth transition from historical emissions to scenario emissions. Scenarios where bunkers emissions have not been removed before downscaling the post 2050 emissions include full bunkers emissions.

Figure 5 shows the effect of harmonization on aggregate Kyoto GHGs emissions for the 5 regions used for the RCPs. The harmonization factors are the same for all countries in a region as the same historical data are used during the downscaling to calculate the initial emissions intensities and for the harmonization during the combination of scenario data with historical
data.

The harmonized SSPbe country data and downscaled SSPv2 data are combined with historical data such that historical data takes precedence over scenario data where both are available. The last year with historical data is 2017. The combined time series are used in the downscaling process of emissions data.

### 4.5 Downscaling of RCP and SSPv2 emissions data

RCP and SSPv2 downscaling uses the IPAT based convergence downscaling with exponential convergence of emissions intensities as introduced in Section 2.3 for all gases and sectors. The parameters are the same for RCP and SSPv2. The convergence years are set for each SSP individually but with no regional variation. Convergence years are shown in Table 4.

**Table 4.** Convergence years for IPAT based convergence downscaling for the different SSPs.

| Scenario | SSP 1 | SSP 2 | SSP 3 | SSP 4 | SSP 5 |
|---|---|---|---|---|---|
| Year | 2150 | 2200 | 2300 | 2300 | 2150 |

For the RCPs we use the SSPbe country data as downscaling key while for SSPv2 we use the data from the scenarios downscaled to country-level (Section 4.3).

To illustrate the influence of the downscaling method on the results, we complement our standard results using IPAT downscaling with exponentially converging emissions intensities (IE, see Section 2.3) with comparison datasets using IPAT downscaling with constant relative emissions intensities (IC, see Appendix C1), and constant shares downscaling (CS, see Section 2.2 and Appendix C2), which is independent of the socio-economic scenarios.

Earth System
**Science**
**Data**

**Figure 5.** Harmonization of emissions scenarios to historical data. The harmonization factor is phased out until 2050. The regions shown here are built from downscaled data to harmonize the region definitions of the SSPv2 scenarios to the RCP regions. As the region definitions of some IAMs do not cover all countries the aggregate data is in some cases lower than the historical data (see e.g. Middle East and Africa region). Harmonization factors are similar for all RCPs as the IAM groups use the same historical data for all their scenarios. Harmonization factors for individual gases, especially fluorinated gases, can be larger (or in case of $CO_2$ smaller). See SI, Section 2.2 for details.





We downscale each RCP SSP combination from Table 2 for all three GDP country model groups (IIASA, PIK and OECD)
and each SSPv2 scenario from Table 1 for all available IAM implementations (see Appendix A1 for details). Each of these
scenarios are available in two versions, one where the scenarios have been corrected for bunkers emissions and one where they
have not been corrected.

## 4.6   Combination with historical data

The downscaled RCP and downscaled SSPv2 data are combined with historical emissions data such that historical data takes
precedence over scenario data where both are available. The last year with historical data is 2017. The scenario data are
harmonized to interpolated 2017 historical data as described in Section 4.4.

## 5   Results

Here, we discuss and present the data for some key countries, selected from all five regions. From the Asia regions we selected
China, Afghanistan, and South Korea, to represent the diverse economical situations present in the region. Afghanistan is
an extreme case not only in the Asia region but globally, as the SSP basic elements project very high GDP[3] growth rates
(Figure 6). From Latin America we select Brazil and Guatemala, which differ substantially in historical emissions intensity
and economical development under the SSPs. The Middle East and Africa region is represented by South Africa, a country with
relatively high GDP per capita, and Ethiopia with one of the world's lowest GDP per capita values but very high economical
growth in recent years. For the OECD we selected three countries: two major world economies, the USA and Great Britain,
which have high GDP per capita, with the USA having twice the UK's emissions per unit of GDP; the third country is Bulgaria,
with roughly half of the UK's GDP per capita. The Reforming Economies region is represented by its main economical power
Russia, and by Uzbekistan, which has a GDP per capita of less than one third of the Russian GDP per capita. For these countries
we present selected downscaled scenarios to highlight some of the factors influencing the downscaling results.

The final GDP scenarios are displayed in Figures 6 and 7. The SSPv2d GDP pathways are very similar for all IAM groups
as they are all based on the same OECD country data. The SSPbe data from IIASA and PIK vary substantially from the OECD
and SSPv2d data for several countries and scenarios.

Figures 8 and 9 show the resulting country pathways for RCP 2.6, aggregate Kyoto GHGs, and IPAT downscaling with
exponential convergence. Results for individual gases and all RCPs can be found in the SI in Section 2.4. Figures 10 and 11
compare the exponential IPAT results with other downscaling methods for RCP 2.6, SSP 2. Results for individual gases and an
additional RCP SSP combination (RCP 6.0, SSP 5) can be found in the SI in Section 2.5.

Influence of the socio-economic scenarios on downscaled emissions is high where the socio-economic scenarios and / or
historical emissions intensity are diverse within a region. Where they are similar, the resulting country emissions pathways
are similar. We have selected countries which differ in at least one of these indicators for the example plots shown, and
consequently all regions show some differentiation. In relatively homogeneous regions like the OECD the differentiations are

---

[3]All references to GDP in this section are referencing purchasing power parity (PPP) adjusted GDP.



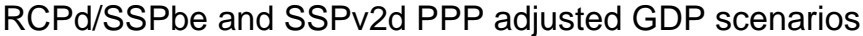

Figure 6. GDP projections for all SSPs (basic elements and downscaled SSPv2). All scenarios harmonized to historical values. Countries from OECD and Asia regions. The SSP basic elements data from PIK and IIASA vary substantially from the OECD data and the OECD based SSPv2 data for some countries and SSPs. GDP growth varies substantially between scenarios.

**Figure 7.** GDP projections for all SSPs (basic elements and downscaled SSPv2). All scenarios harmonized to historical values. Countries from Latin America, Reforming Economies, and Middle East and Africa regions. The SSP basic elements data from PIK and IIASA vary substantially from the OECD data and the OECD based SSPv2 data for some countries and SSPs.





small and comparable to the spread of scenarios from different modeling groups, while for regions with large differences in historical emissions intensity and / or GDP growth rates, the country pathways are show strong variations. The most prominent example in the figures is Afghanistan, where high GDP growth and converging emissions intensities lead to 2100 negative emissions, which are more than twice the current positive emissions. Relatively developed countries in the same region have much smaller negative emissions relative to current emissions levels. We have to note here that this is a downscaling study, not

an equity study. There is no implication of fairness in the resulting pathways.

The influence of the downscaling method is most prominent for regions with high economic differentiation as well. However, especially for the Asia region, the differences between methods are more visible than the differences between scenarios. The major influence of the GDP growth rates is clearly visible from Figures 10 and 11: constant shares downscaling, which does not take the GDP scenarios into account, differs strongly from the other methods, which use GDP data. When comparing pathways

with convergence (BIE) with pathways without convergence (BIC), the influence of convergence of emissions intensities is visible as well, but less prominent.

Figures 12 and 13 show the influence of the correction of scenarios for bunkers emissions on the example countries EU (Figure 12) and Ethiopia (Figure 13). As bunkers emissions are distributed to the regions based on historical emissions shares the influence is much larger for the EU than for Ethiopia. While the absolute emissions difference is higher for high emissions

scenarios, considering bunkers emissions can be decisive for net-negative or net-positive emissions in high mitigation scenarios (Figure 12). Figures for the other example countries can be found in the SI in Section 2.6.

## 6   Discussion and Limitations

The main challenge for a downscaling methodology is to produce sensible results in regions with diverse economical situations, especially in scenarios with strongly decreasing or even negative emissions. The established IPAT with exponential

convergence method does not work for numerical reasons and is problematic because it produces early negative emissions for countries with low historical emissions intensities, no matter if this is due to poverty or low emissions technologies. We opted to converge emissions intensities before the transition to negative emissions, which alters the concept of convergence. Generally, a downscaling process always needs several assumptions that influence the final data. These assumptions impact both downscaling results and results of studies based on the downscaled data. Thus, downscaled data also has to be used with

caution and keeping the assumptions made in mind. In the following, we list the main limitations of our approach and their impacts on the resulting emissions pathways.

The choice of methodology is obviously a main driver of the downscaling results (Figures 10 and 11). We chose the IPAT based convergence downscaling (Section 2.3) as methodology for the main dataset, but also offer downscaled data using the IPAT equation and constant relative emissions intensities (Appendix C1) as well as simple constant share downscaling

(Section C2) for reference.

In all cases, the convergence method (Section 2.3) assumes at least partial convergence of emission intensities within a region. There is not much literature on the convergence of emission intensity of the GDP, while for the energy intensity of the



**Figure 8.** Results for RCP 2.6 and all SSPs for both RCPd and SSPv2d scenarios. Countries from OECD and Asia regions. For the OECD region the differences in country pathways are driven by differences in regional pathways, not by the socio-economic development leading to very similar pathways for the countries in the region. In the Asia region the socio-economic development is more diverse and strongly influences the resulting country emissions pathways. Most prominent example are the high negative emissions for Afghanistan in SSP 5 driven by the very high GDP growth rates (Figure 6). Note: Korea is in the OECD region for the WITCH GLOBIOM model.



**Total Kyoto gases (AR4) excl. LULUCF under RCP 2.6**

**Figure 9.** Results for RCP 2.6 and all SSPs for both RCPd and SSPv2d scenarios. Countries from Latin America, Reforming Economies and Middle East and Africa regions. All regions show strong influences of the diverse GDP growth projections (Figure 7) within the regions.



**Figure 10.** Influence of the downscaling method on country emissions for RCP 2.6, SSP 2. Ranges are calculated over all IAM implementations in SSPv2d. RCPd scenarios are shown as individual lines only (dashed). For the OECD region the influence of the downscaling method is comparable to the influence of the IAM modeling group, but with a clear influence of high GDP growth on emissions in the second half of the century. Countries in the Asia region have a more diverse GDP development (Figure 6) and show much stronger influence of the downscaling method.



**Figure 11.** Influence of the downscaling method on country emissions for RCP 2.6, SSP 2. Ranges are calculated over all IAM implementations in SSPv2d. RCPd scenarios are shown as individual lines only (dashed). For the Latin America region the influence is comparable to the OECD region and mainly visible in the second half of the century (top panels). The Middle East and Africa region shows a strong influence of the downscaling method (bottom panels).

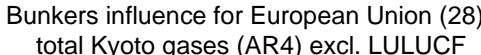

**Figure 12.** Influence of bunkers emissions on results for the European Union. Ranges and median are calculated over all SSPs and all IAM implementations in SSPv2d for each RCP. RCPd scenarios are shown as individual lines only (dashed). Absolute bunkers emissions are high for higher emissions scenarios, however, for low emissions scenarios bunkers emissions can make the difference between net-negative and net-positive emissions.



**Figure 13.** Influence of bunkers emissions on results for Ethiopia. Ranges and median are calculated over all SSPs and all IAM implementations in SSPv2d for each RCP. RCPd scenarios are shown as individual lines only (dashed). Generally the influence of bunkers emissions is much smaller than for the EU, because the global bunkers scenario is split to regions using historical shares which are high for the OECD and REF regions (EU) and low for the Africa region compared with future emission.





GDP studies exist (Liddle, 2010; Markandya et al., 2006). The energy sector is a main driver of greenhouse gas emissions for most countries and therefore energy intensity of the GDP is a major input to the emission intensity of the GDP. Historical data

show that the energy intensity often converges within regions (Markandya et al., 2006), however, this is not true for all regions (Liddle, 2010).

Several studies deal with the convergence of per capita $CO_2$ emissions and come to different conclusions: Stegman and McKibbin (2005) find that when a large cross section of countries is considered there is little evidence for convergence, while there is some evidence of convergence within the OECD region. This is generalized by Panopoulou and Pantelidis (2009) who

find that convergence to different per capita emissions levels exist, a concept they call club convergence. Strazicich and List (2003) find that per capita $CO_2$ emissions levels have converged among 21 industrialized countries, which is confirmed by Romero-Ávila (2008), Jobert et al. (2010), and Chang and Lee (2008). Ordás Criado and Grether (2011) study 166 countries and find convergence within groups of countries (similar income, neighboring, institutional partners) between 1980 and 2000, especially within the EU and OECD regions.

This shows that convergence of emissions intensities within regions is a sensible assumption, but it is important to note that it is an input to the downscaling process and thus the emissions intensities of the downscaled data are an input to and not a result of the process.

As a regional transition to negative emissions and negative emissions intensities has not yet been observed, there is no evidence if convergence is a sensible assumption for pathways with negative emissions. It is not yet clear which technologies will

be used to achieve negative emissions. While for some technologies (e.g., direct air capture) a relation to GDP seems sensible, other technologies like BECCS also depend on national circumstances such as the availability of land to grow energy crops and safe storage options for the captured $CO_2$. More generally, the downscaling methodology considers the emissions intensity per country and gas but does not consider the reasons for high and low emissions intensities. Thus, if a region contains two countries with similar emissions intensities in the harmonization period, the algorithm will create similar emissions intensity

pathways for both countries - neither considering if, e.g., a low emission intensity comes from a low development level or a high share of renewables, nor considering the potential for mitigation technologies (e.g. for BECCS). As our analysis is carried out on national total emissions per gas it is also not taken into account if, e.g., high methane emissions come from agriculture or fugitive emissions from fossil fuel production and handling, which are easier to mitigate than agricultural emissions.

The SSP basic elements results assume very high GDP growth for several developing countries. For Afghanistan the 2100

GDP values are between roughly 16 (SSP 4) and over 110 (SSP 5) times the 2015 value, to give one of the most extreme examples. Countries in the same region that are more developed do not exhibit these kind of growth rates. With (relative) GDP being a driver of emissions in the downscaling model, emissions increase accordingly by a large factor or - in case of strong mitigation pathways - decrease to minus several 100 percent of the 2015 emissions. The resulting absolute emissions pathways seem highly unrealistic but are merely a result of the strong GDP growth in the SSP basic elements results.

There are several sources for uncertainty in the data presented here, e.g., the uncertainty in historical emissions and GDP data. But the most important source of uncertainty is that we are using scenarios that project GDP and emissions development 90 years into the future under given broad storylines. These scenarios are based on several assumptions and can only model





idealized economical and technological developments. Some of the scenarios use technologies not yet proven to be applicable on large scale (BECSS), technologies deemed too dangerous to be used by several countries (nuclear), assume that we solve

the problem of high variability in availability of renewable energy sources, et cetera. On the other hand the models cannot anticipate new and still unknown technologies, which might solve the problem in ways not imaginable today. Furthermore, the emissions reductions in IAMs mainly come from technological change in energy production. The energy intensity of the GDP is very similar between baseline and mitigation scenarios (see, e.g. Figure 3 of Peters et al., 2017). Thus, an important possibility for emissions reduction - the reduction of energy use - is only considered partly by IAM scenarios. The underlying

population and GDP projections do not model crises as the 2008 financial crises or the 2020 global COVID-19 pandemic and their impact on lives and economy.

Most IAM scenarios have been created using some kind of cost optimization routine and therefore assume costs to be the driving factor of economic decision making - just steered by parameters such as a carbon tax or an emissions cap. They thus assume an idealized version of the current economical system where cost-optimal decisions are made on a rational basis.

Essentially, IAM scenarios assume no revolutionary changes (be it of technological or societal nature) but rather a continuation of our current system with some modifications to reduce GHG emissions. While IAMs are one of the main tools to generate and assess socio-economic scenarios to mitigate climate change, their usefulness is not undisputed in the scientific community (see, e.g. Jewell and Anderson, 2019). Whenever the country-level scenarios presented here are used, the limitations and assumptions of the downscaling process as well as the underlying models have to be taken into account.

## 7 Conclusions

The country resolved downscaled RCP and downscaled SSPv2 scenarios we present here allow for climate policy analysis in terms of RCP GHG forcing scenarios and SSP socio-economic storylines on a per country basis. While we treat the "IE" method with converging emissions intensities as our main dataset and use the others for reference, users can opt for more conservative assumptions using the datasets which employ constant shares (CS) and constant relative emissions intensities

(IC) downscaling. Earlier version of the scenarios presented here have been used in several studies (Meinshausen et al., 2015; Robiou du Pont et al., 2016; du Pont et al., 2016; Robiou du Pont and Meinshausen, 2018) and are used by the climate policy assessment of the Climate Action Tracker (CAT: Climate Analytics and New Climate Institute, 2020). With this paper we make the data publicly available and describe the used methodology in detail. We hope that this enables a broader use of the data.

## 8 Data availability

All datasets produced for this manuscript are available for download at https://doi.org/10.5281/zenodo.3638137 (Gütschow et al., 2020). Each dataset comes in a csv file. The file name is constructed as follows: <Source><Bunkers><Downscaling>.

The <Source> flag indicates which input scenarios were used.



**PMRCP** RCP scenarios downscaled using the SSPbe: emissions and socioeconomic data; scenarios are available both harmonized to historical data and non-harmonized.

**PMSSP** Downscaled SSPv2 scenarios: emissions and socioeconomic data; scenarios are available both harmonized to historical data and non-harmonized.

the <Bunkers> flag indicates if the input emissions scenarios have been corrected for bunkers emissions before downscaling to country level or not. The flag is "B" for scenarios where emissions from bunkers have been removed before downscaling and "" (empty) where they have not been removed. We recommend using datasets where bunkers emissions have been removed

before downscaling.

The <Downscaling> flag indicates the downscaling technique used.

**IE** Convergence downscaling with exponential convergence of emissions intensities and convergence before transition to negative emissions. (see Section 2.3, default downscaling method)

**IC** Regional emission intensity growth rates for all countries. (see Appendix C1)

**CS** Constant emission shares as a reference case independent of the socio-economic scenario. (see Appendix C2)

All files contain data for all countries and variables, all scenarios (RCPd or SSPv2d) both harmonized and unharmonized. We recommend the use of the "BIE" dataset as a default.

More information on the data structure of the files is a available in the data description in the data repository (Gütschow et al., 2020).

**Appendix A: Scenario details**

**A1 SSPv2 scenarios**

Most IAMs have run all RCP SSP combinations, but the highest and lowest RCP forcing levels were not attained by all models for all SSPs. Additionally some models have not simulated all SSPs. Table A1 gives an overview over the available models for each RCP SSP combination.

**Appendix B: Methodological details**

**B1 PPP conversion of SSP socio-economic data**

Recent historical GDP data in purchasing power parity (PPP) corrected form uses the ppp data from the 2011 revision of the International Comparison Program (ICP, World Bank, 2014). It is published in units of 2011 International Dollar (or Geary-Khamis-Dollar, GKD). Data in 2011GKDs is ppp corrected using the 2011 ICP and the exchange rate to the US Dollar is set





**Table A1.** SSP IAM implementations available in the SSPv2 database. Model names are abbreviated using the first character. Illustrative marker scenarios are marked by bold italic letters. There are no RCP 8.5 scenario implementations as no SSP IAM baseline shows forcing levels above RCP 8.5.

|            | SSP 1              | SSP 2              | SSP 3          | SSP 4        | SSP 5           |
|------------|--------------------|--------------------|----------------|--------------|-----------------|
| Baseline   | *I*, M, A, G, R, W | I, *M*, A, G, R, W | I, M, *A*, G, W | I, A, *G*, W | I, A, G, *R*, W |
| RCP 8.5    | —                  | —                  | —              | —            | —               |
| RCP 6      | —                  | I, *M*, A, G, R, W | I, M, *A*, W   | I, *G*, W    | I, A, G, *R*, W |
| RCP 4.5    | *I*, M, A, G, R, W | I, *M*, A, G, R, W | I, M, *A*, W   | I, A, *G*, W | I, A, G, *R*, W |
| $3.4W/m^2$ | *I*, M, A, G, R, W | I, *M*, A, G, R, W | I, M, *A*, W   | I, A, *G*, W | I, A, G, *R*, W |
| RCP 2.6    | *I*, M, A, G, R, W | I, *M*, A, G, R, W | —              | I, A, *G*, W | A, G, *R*       |
| $1.9W/m^2$ | *I*, M, A, G, R, W | I, *M*, A, G, R    | —              | W            | G, *R*          |
| Marker     | IMAGE              | MESSAGE            | AIM/GCE        | GCAM4        | REMIND          |

to be 1 USD = 1 2011GKD in 2011. The SSP basic elements data are published in 2005 international dollars and thus based on the 2005 ICP revision (World Bank, 2008) making conversion between the two units necessary.

The conversion consists of two steps: all countries but the US have changed ppp factors that relate the countries purchasing power to the US purchasing power; additionally the purchasing power of the reference currency (USD) changed between the revisions requiring a further correction. The World Development Indicators (WDI, The World Bank, 2019b, a) provide ppp

factors used to convert data in international dollars into local currency units (LCA) for both 2005 (indicator PA.NUS.PPP.05, short PPP05) and 2011 (PA.NUS.PPP, short PPP11). These can be used to correct for the different PPP values. To correct for the reference currency we use the GDP inflation (indicator NY.GDP.DEFL.KD.ZG, short INFL) for the US to correct for USD inflation. The full conversion is

$$\text{GDP}_c^{2011\text{GKD}} = \text{GDP}_c^{2005\text{GKD}} \frac{\text{PPP11}_c}{\text{PPP05}_c} \prod_{y=2006}^{2011} \text{INFL}_{\text{USA}}(y), \tag{B1}$$

for country $c$.

For some countries we have to make exceptions from this conversion rule. For Belarus and Mauritania the 2005 PPP factors are very high leading to very high conversion factors and results not in line with historical GDP data in 2011 GKD. For these countries we use the 2005 value given in the PPP11 time series. The same is done if the PPP05 factor is not available. This is the case for Aruba and Eswatini. If a country is not present in the PPP11 time series we use data from the Penn World Table

version 9.1 (PWT: Feenstra et al., 2019, 2015). We multiply the market exchange rate (MER, "xr" in PWT) for the country with the price level of the GDP ("pl_gdpo" in PWT) to obtain a PPP time series. This is used for Djibouti and Syria. If no PPP data are available only the USD inflation is used. This is used for Cuba, French Polynesia, New Caledonia, Puerto Rico, and Somalia. for most of these countries no historical GDPPPP data are available.





This transformation is not strictly needed to create our dataset, as we only use the growth rates of the GDPPPP scenarios, however, the transformed time series give a good indication on how large the discrepancy between historical GDPPPP data and the scenario data is. Except for a few small countries the discrepancy is small.

## Appendix C: Methodological variants

### C1   Downscaling with constant relative emission intensity

To demonstrate the influence of convergence we also created a dataset in which the regional emission intensity growth rates are used for all countries of a region. To achieve that, the emission intensity is held constant at the value of the harmonization year for the temporary emission intensity pathway:

$$\mathrm{EI}_c(y) = \mathrm{EI}_c(y_h). \tag{C1}$$

See also Figure C1. While methodologically very similar to the exponential convergence downscaling this is actually not convergence downscaling, but a form of external input based downscaling.

### C2   Constant shares downscaling

As a control case for the influence of socioeconomic data on the downscaling we also created a dataset not using GDP data at all. We downscaled regional scenarios using historical country shares

$$\mathrm{S}_{\mathcal{C}}(y_0) = \mathrm{E}_{\mathcal{C}}(y_0)/\mathrm{E}_{\mathbb{R}}(y_0) \tag{C2}$$

throughout the whole scenario time frame. Emissions are calculated using

$$\mathrm{E}_{\mathcal{C}} = \mathrm{S}_{\mathcal{C}}(y_0)\mathrm{E}_{\mathbb{R}}. \tag{C3}$$

## Appendix D: Region definitions

For the RCPs we use the definitions of the R5 regions as presented on the RCP database homepage (IIASA, 2009). The SSPs use the updated R5.2 region definitions which are availabel on the SSP database homepage (IIASA, 2018). However, the exact regions differ by IAM, as they have to be created from sums of native IAM regions which differ by IAM. We thus use model specific region definitions based on an Excel document formerly available on the SSPDB homepage checked against (and slightly corrected by) the regions definitions presented in the SSP model documentation (SSP, 2015).

*Author contributions.* JG designed the study, carried out the downscaling process and led the manuscript writing process. All authors discussed the methodology and results and contributed to the manuscript.



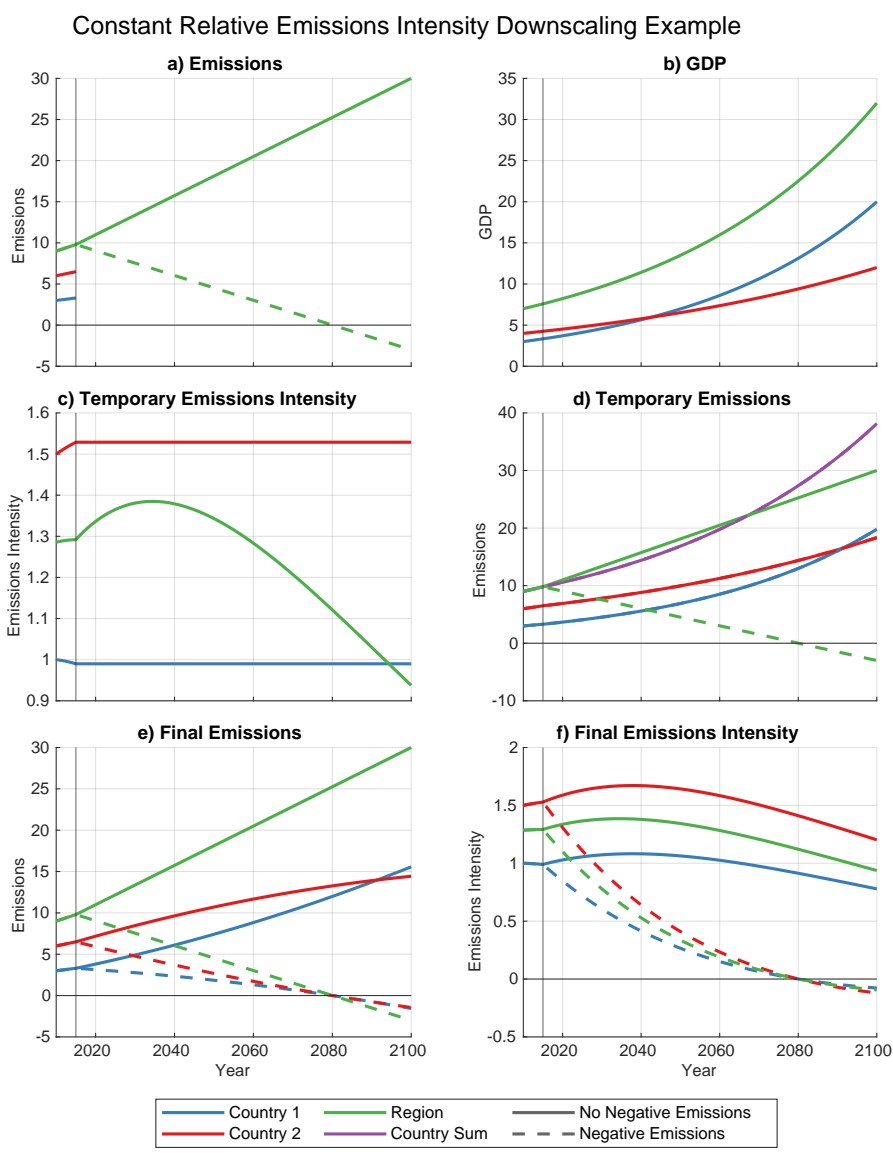

**Figure C1.** Steps of external input based downscaling of regional emissions data using the IPAT equation and country GDP data explained using a two country region. We use the assumption of constant relative emissions intensities to enable the use of GDP as an external input for emissions downscaling. Regional emission data are given for the whole time period while for the countries only historical data are available (Panel a). GDP data are given for both countries and the region (Panel b) for the whole time period. In the first step temporary emission intensity pathways for the countries are calculated using a constant extrapolation of historical values (2015) (Panel c). Multiplication with the given GDP time series creates temporary emissions time series. These do not sum up to the regional value (see Panel d) and have to be scaled to the regional value (results in Panel e). This also changes the emissions intensities (Panel f)



*Competing interests.* We have no competing interests to declare.

*Acknowledgements.* JG and AG acknowledge support by the German Federal Ministry for the Environment, Nature Conservation and Nuclear Safety (16_II_148_Global_A_IMPACT). JG and AG further acknowledge support by the German Federal Ministry of Education and Research (01LS1711A).





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
