# Peer review of "Country resolved combined emission and socio-economic pathways based on the RCP and SSP scenarios"

_Earth System Science Data, 2020_

## Referee Comment (RC1) · Anonymous Referee #1 · 8 Jul 2020

This paper describes the production of a country-level dataset of greenhouse gas emissions and socioeconomic pathways, by downscaling relevant variables from existing regional-scale scenarios based on the RCPs and the SSPs. The dataset has value, and the methods used are sensible. However, the presentation is lengthy, with expansive background information (occasionally in the methods section) that sometimes does not seem essential for communicating the work, e.g. most of the text on SRES. In my opinion, removing such text would improve clarity. Overall, the work and paper are of good quality.

---

## Referee Comment (RC2) · Anonymous Referee #2 · 9 Dec 2020

This manuscript summarizes a dataset with country-resolved emission and socioeconomic pathways under RCP-SSP scenarios. It comprehensively introduces the background, methods and shows the example results. The data is also publicly available and well documented. I enjoyed reading the manuscript as well and particularly liked the introduction that gives the reader a really good overview of the scenarios used in the study.

The only concern I have is the possible jargons for broad readership. As this paper is really lengthy, it is easy to get lost of some abbreviations. I would recommend add a list of abbreviations and definitions for the key elements in the paper in the appendix.

[Figure]

Otherwise, this is a nice and timely paper that well fits the scope of ESSD.

A minor question, in Fig 1 panel A. What's the dashed line?

---

## Author Comment (AC1) · 4 Jan 2021

Dear Referees,

thank you very much for reviewing this long paper. And thank you also for your positive feedback. I will incorporate your comments into a revised version and check where the paper can be shortened without removing content. I will add a list of abbreviations and definitions. Regarding the question on Figure 1. The dashed line in panel a is the negative emissions pathway also displayed in panel d. I will add that to the legend.

Best regards, Johannes Gütschow

---

## Author Response (AR1)

**Author response essd-2020-101**

**1. Comments from Referees**

- **Anonymous Referee #1**
  This paper describes the production of a country-level dataset of greenhouse gas emissions and socioeconomic pathways, by downscaling relevant variables from existing regional-scale scenarios based on the RCPs and the SSPs. The dataset has value, and the methods used are sensible. However, the presentation is lengthy, with expansive background information (occasionally in the methods section) that sometimes does not seem essential for communicating the work, e.g. most of the text on SRES.
  In my opinion, removing such text would improve clarity. Overall, the work and paper are of good quality.

- **Anonymous Referee #2**
  This manuscript summarizes a dataset with country-resolved emission and socioeconomic pathways under RCP-SSP scenarios. It comprehensively introduces the background, methods and shows the example results. The data is also publicly available and well documented. I enjoyed reading the manuscript as well and particularly liked the introduction that gives the reader a really good overview of the scenarios used in the study.
  The only concern I have is the possible jargons for broad readership. As this paper is really lengthy, it is easy to get lost of some abbreviations. I would recommend add a list of abbreviations and definitions for the key elements in the paper in the appendix.
  Otherwise, this is a nice and timely paper that well fits the scope of ESSD.
  A minor question, in Fig 1 panel A. What's the dashed line?

**2. Author's response**

Dear referees, dear editor.

Thank you very much for reviewing / handling this long paper. And thank you also for your positive feedback. We have incorporated your comments into a revised version and removed a few sentences to give more focus to the methods section. As comments on the length of the paper and the content were mixed among the referee reports we have only removed few sentences and kept the content mainly as it was.
We have added tables with acronyms and definitions used in the paper.

Best regards,
Johannes Gütschow

**3. Author's changes in manuscript.**

- We have corrected several typos.
- At some points we have reformulated sentences for easier understanding by readers.
- In the methods section we have rewritten parts of the text referring to the SRES scenarios to clarify where we base our approach on approaches to downscale SRES data. We have also removed some sentences that were not necessary for the methods section. (in response to ref #1)
- We have added a section with several tables listing relevant definitions and abbreviations in the appendix. (in response to ref #2)
- We have added the dashed line in Figure 1, panel a to the legend. (in response to ref #1).
- We have also changed panel headings and legend entries in Figures 1 and 2 to improve clarity.
- In Figure 4 we have adjusted the colors for better visibility.

A marked-up manuscript is attached to this document.

[revised manuscript text omitted]